# Hair Cortisol and DHEA-S in Foals and Mares as a Retrospective Picture of Feto-Maternal Relationship under Physiological and Pathological Conditions

**DOI:** 10.3390/ani12101266

**Published:** 2022-05-14

**Authors:** Aliai Lanci, Jole Mariella, Nicola Ellero, Alice Faoro, Tanja Peric, Alberto Prandi, Francesca Freccero, Carolina Castagnetti

**Affiliations:** 1Department of Veterinary Medical Sciences (DIMEVET), University of Bologna, Via Tolara di Sopra 50, Ozzano dell’Emilia, 40064 Bologna, Italy; aliai.lanci2@unibo.it (A.L.); jole.mariella2@unibo.it (J.M.); alice.faoro2@unibo.it (A.F.); francesca.freccero2@unibo.it (F.F.); carolina.castagnetti@unibo.it (C.C.); 2Department of Agricoltural Food, Environmental and Animal Science (DI4A), University of Udine, Via delle Scienze 206, 33100 Udine, Italy; tanja.peric@uniud.it (T.P.); alberto.prandi@uniud.it (A.P.); 3Health Science and Technologies Interdepartmental Center for Industrial Research (CIRI-SDV), University of Bologna, Via Tolara di Sopra 50, Ozzano dell’Emilia, 40064 Bologna, Italy

**Keywords:** neonatal foal, mare, pregnancy, prenatal, hair, hormones, ratio, hypothalamus pituitary adrenal axis, allostasis, biomarkers

## Abstract

**Simple Summary:**

Hair hormone concentrations represent an attractive alternative for studies focusing on the prenatal period because a single measurement from hair collected at birth could provide data from a wide but datable prenatal period. Our aim is to evaluate cortisol, dehydroepiandrosterone-sulfate concentrations and their ratio in the trichological matrix of mares and foals to assess maternal and fetal hypothalamus-pituitary-adrenal axis activity in relation to the clinical condition of the neonate and the housing place at parturition. Mare’s endocrine setting influences hormone concentrations detected in foal’s hair, and foals classified as sick at birth have different hormone concentrations than healthy ones. On the other hand, hormone concentrations detected in the hair of mares hospitalized for attended parturitions do not reflect a different hypothalamus-pituitary-adrenal axis activity from those foaled at the breeding farm. Although the specific role of cortisol and dehydroepiandrosterone-sulfate in foals and mares under pathological conditions remains under investigation, this study opens up the use of neonate’s hair as an attractive non-invasive retrospective calendar for the assessment of prenatal adrenal activity in the equine species.

**Abstract:**

Equine fetal hair starts to grow at around 270 days of pregnancy, and hair collected at birth reflects hormones of the last third of pregnancy. The study aimed to evaluate cortisol (CORT) and dehydroepiandrosterone-sulfate (DHEA-S) concentrations and their ratio in the trichological matrix of foals and mares in relation to their clinical parameters; the clinical condition of the neonate (study 1); the housing place at parturition (study 2). In study 1, 107 mare-foal pairs were divided into healthy (group H; *n* = 56) and sick (group S; *n* = 51) foals, whereas in study 2, group H was divided into hospital (*n* = 30) and breeding farm (*n* = 26) parturition. Steroids from hair were measured using a solid-phase microtiter radioimmunoassay. In study 1, hair CORT concentrations measured in foals did not differ between groups and did not appear to be influenced by clinical parameters. A correlation between foal and mare hair CORT concentrations (*p* = 0.019; r = 0.312, group H; *p* = 0.006; r = 0.349, group S) and between CORT and DHEA-S concentrations in foals (*p* = 0.018; r = 0.282, group H; *p* < 0.001; r = 0.44, group S) and mares (*p* = 0.006; r = 0.361, group H; *p* = 0.027; r = 0.271, group S) exists in both groups. Increased hair DHEA-S concentrations (*p* = 0.033) and decreased CORT/DHEA-S ratio (*p* < 0.001) appear to be potential biomarkers of chronic stress in the final third of pregnancy, as well as a potential sign of resilience and allostatic load in sick foals, and deserve further attention in the evaluation of prenatal hypothalamus-pituitary-adrenal (HPA) axis activity in the equine species. In study 2, hormone concentrations in the hair of mares hospitalized for attended parturition did not differ from those that were foaled at the breeding farm. This result could be related to a too brief period of hospitalization to cause significant changes in steroid deposition in the mare’s hair.

## 1. Introduction

The hypothalamus-pituitary-adrenal (HPA) axis is a regulatory system whose goals include mounting a metabolic response to stressful conditions to guarantee survival. On stressful stimulation, the hypothalamus releases a corticotropin-releasing hormone into the pituitary portal system to stimulate the secretion of adrenocorticotropin (ACTH) by pituitary corticotrophs. The main target for ACTH is the zona fasciculata of the adrenal gland cortex to produce glucocorticoids, in particular cortisol (CORT) [1]. Dehydroepiandrosterone (DHEA) is another marker of adrenocortical activity in response to stressful events. DHEA and its sulfate metabolite (DHEA-S) are produced in the zona reticularis of the adrenal glands and have opposite functions to CORT [2].

Increased plasma CORT concentrations have been found in pregnant mares with acute medical and surgical abdominal disease. Endotoxemia is often associated with this condition, and endotoxins are likely a key stimulus for HPA axis activation [3]. In a recent study, concentrations of DHEA/DHEA-S have been investigated in mares undergoing experimentally induced ascending placentitis [4]. HPA axis dysregulation also seems to be a frequent problem in critically ill foals. ACTH and CORT concentrations were increased in septic foals, and foals with the highest plasma ACTH concentrations were more likely to die [5,6,7,8]. These results suggest that excessive hypothalamic and pituitary responses are either risk factors for death or reflect the terminal stage of the disease. In foals affected by Neonatal Maladjustment Syndrome, characterized by neuronal failure resulting from hypoxia and ischemia of the brain that occurs before, during, or after parturition, plasma DHEA concentrations were higher than in healthy ones at 24 and 48 h of life [9]. Calculating the CORT/DHEA(S) ratio can be considered the correct approach to studying the relative abundance of the two hormones and determining which is preferentially produced. As CORT and DHEA(S) serve interconnected but largely opposing functions, there has been growing interest in the potential value of examining the two as a ratio, reflecting their proportional levels [2].

Plasma, salivary, fecal, and urine glucocorticoid concentrations reflect acute stress, but they cannot represent a long-term retrospective integrative HPA axis activity as hair hormones do [10]. Many factors, e.g., the restraint techniques used during venipuncture or the time-to-time variability, make the interpretation of blood hormone data challenging, making it difficult to study the effect of environmental factors or pathological conditions on glucocorticoid production.

The hormone measurement in hair samples provides a “retrospective picture” of hormone incorporation from plasma and accumulation over a time-lapse [10]. While hair CORT concentrations have already been evaluated in several species [10], only a few recent studies have analyzed hair CORT concentrations in horses. Higher hair CORT concentrations have been found in healthy newborn foals and 30–60-day-old foals compared to healthy adult horses [11,12]. In those foals, hair CORT concentrations are shown to decrease after birth [11] and not be influenced by environmental factors such as temperature, rainfall, and day length [12]. In the study performed by Duran et al. on horses undergoing acute stress [13], hair CORT concentrations were higher one and two months after surgical intervention. Hair CORT concentrations were also increased in horses with pituitary pars intermedia dysfunction compared with healthy ones [14]. However, lower hair CORT concentrations have been reported in horses with a severe squamous gastric disease when compared to horses without gastric ulcers [15]. Although the CORT/DHEA ratio in equine hair has been previously described to evaluate the influence of different housing conditions [16] or seasonality [17], there is still a lack of information on its biological relevance in horses, especially in pregnant mares hospitalized for assisted delivery.

Equine fetal hair starts to grow at around 270 days of pregnancy [18,19], so hormone concentrations measured in the hair samples collected at birth reflect their accumulation in the last third of pregnancy. The final 15 days of pregnancy are not represented by hormones accumulated in the hair similar to what is observed in humans, the section of hair located beneath the scalp is not collected because hair is always shaved close to the skin and not plucked. This section of hair represents a lag time of 2 weeks since the hair grows 1 cm per month [20]. Hair hormone concentrations represent an attractive alternative for studies focusing on the final third of pregnancy because it avoids invasive blood collection from the fetus. A single measurement from hair collected at birth will provide data from a wide but datable prenatal period. In the pregnant mare, circulating CORT peaks immediately prior to delivery and would not affect the concentration of CORT in fetal hair at birth [21].

The present study aimed to evaluate CORT and DHEA-S concentrations and their ratio in the trichological matrix of foals and mares in relation to the selected mare’s reproductive characteristics (age, parity, gestational length); selected foal’s physical/clinical characteristics (sex, weight, outcome); clinical condition of the neonate (healthy vs. sick; study 1); housing place at parturition (hospital vs. breeding farm; study 2). The present investigation is based on the hypothesis that (i) hormone concentrations detected individually in foal’s hair are influenced by the endocrine setting of the respective mare; (ii) foals classified as sick at birth have different hormone concentrations than healthy ones; (iii) hormone concentrations detected in the hair of mares hospitalized for attended parturition reflect different HPA-axis activity from those that foaled at the breeding farm.

## 2. Materials and Methods

### 2.1. Population

One-hundred seven mares with their respective foals hospitalized for attended parturition or admitted immediately after foaling because of the foal’s clinical condition to the Perinatology and Reproduction Unit (Equine Clinical Service, Department of Veterinary Medical Sciences) of the University of Bologna during 2017–2020 breeding seasons were included in the study.

Fifty mares were hospitalized at about 310 days of pregnancy because the owners requested an attended parturition. They were housed in separate wide straw-bedded boxes, fed hay ad libitum and concentrates twice a day and were allowed to go to pasture during the day. A clinical evaluation was performed at admission, including complete blood count, serum biochemistry and transrectal ultrasonography for pregnancy monitoring. Based on the clinicians’ judgment, transabdominal ultrasonography was performed when indicated. Subsequently, mares were clinically evaluated twice a day and by ultrasonography every 5–10 days until parturition. A cervical swab was performed to obtain a bacterial culture when an increase in combined thickness of the uterus and placenta (CTUP) and/or purulent/serosanguineous vulvar discharge were observed. After delivery, a macroscopic examination of the placenta was performed in all mares, and samples of the placenta were collected for histopathologic examination in mares with a high-risk pregnancy.

High-risk pregnancy was defined as a history of premature udder development/lactation, increase of CTUP, purulent/serosanguineous vulvar discharge or mares’ systemic illness or based on gross and histopathological examination of the placenta [22,23]. Mares with high-risk pregnancies were treated based on the clinicians’ judgment.

A complete clinical evaluation, including complete blood count and serum biochemistry, was performed at birth in all the 50 foals born at the Unit. Blood culture and arterial blood gas analysis were performed when indicated based on the clinicians’ judgment.

Fifty-seven pairs of mare and foal were admitted to the hospital immediately after parturition because of the foal’s clinical condition. All the sick foals received a complete clinical evaluation, including blood culture, complete blood count, serum biochemistry, arterial blood gas analysis and serum IgG determination. Both mares and foals were monitored throughout the hospitalization period by a clinical examination performed every 6–12 h, depending on the severity of existing conditions.

In study 1, the population was divided into healthy foals (group H) and sick foals (group S) based on history and clinical evaluation performed at birth/admission.

Foals born from normal pregnancy were classified as healthy when they had an Apgar score ≥ 9 [24] and a normal clinical evaluation during hospitalization, including an IgG serum concentration > 800 mg/dL at 12 h of life.

Foals born at the Unit or referred after birth were classified as sick only when affected by a disease related to the intrauterine life. Sick foals were classified as affected by Hypoxic-Ischemic Encephalopathy (HIE) when the hypoxic insult was evident based on history and clinical signs, especially those of neurologic dysfunction [25], and exclusion of other neurologic diseases such as meningitis or trauma. Typical historical events included high-risk pregnancy, and common clinical signs included loss or absence of suckle reflex, inappropriate teat-seeking behavior, dysphagia, hyperreactivity, and weakness [26]. Foals affected by HIE with evidence of dystocic parturition were excluded. Foals with the same clinical presentation but without evidence of a hypoxic insult were classified as affected by Neonatal Syndrome (NS) [27]. Foals were defined as premature when born prior to 320 days of gestation and dysmature when born after 320 days but with immature physical characteristics (e.g., low body weight and inability to maintain body homeostasis) [23]. Foals were classified as affected by flexural deformity when a severe deviation of a limb in the sagittal plane was present and expressed as persistent hyperflexion of a joint region [28]. Extra-uterine diseases excluded were: sepsis acquired after birth, neonatal isoerythrolysis, hemorrhagic shock, meconium retention, uroperitoneum and umbilical remnants diseases.

In study 2, only group H foal-mare pairs were considered to assess the effect of the housing place at parturition on hair hormone concentrations. Mares of group H were divided into hospital parturition (group HH) and breeding farm parturition (group HB).

### 2.2. Clinical Data and Sample Collection

The following data were recorded for each mare: breed, age (years), parity, type of pregnancy (normal/high-risk), prepartum treatment (yes/no), gestation length (days) and pre-parturition housing place (hospital/breeding farm).

The following data were recorded for each foal: breed, sex, weight (kg), Apgar score at birth (only in foals born at the Unit), age at admission (days), diagnosis and outcome.

Using a Heiniger^®^ clipper (Heiniger, Herzogenbuchsee, Swiss), hair was sampled from the sternal region in foals and dorsal cervical region in mares within 24 h from birth/admission/parturition by shaving close to the skin. The samples were stored in the dark at room temperature inside a paper envelope and delivered to the Laboratory of Veterinary Physiology and Reproductive Pathophysiology (Department of Agricultural, Food, Environmental and Animal Science) of the University of Udine, where the analyses were performed.

### 2.3. Hormone Analysis

Hair samples were prepared for steroid assay as described by Probo et al. [29]. Briefly, 40 mg of hair samples were washed in 3 mL isopropanol to ensure the removal of any steroids on their surface. Steroids from hair were extracted with methanol.

CORT [30] and DHEA-S [29] concentrations were measured using a solid-phase microtiter radioimmunoassay (RIA) assay. Briefly, a 96-well microtiter plate (OptiPlate, Perkin-Elmer Life Science, Boston, MA, USA) was coated with goat anti-rabbit γ-globulin serum diluted 1:1000 in 0.15 mM sodium acetate buffer (9 pH) and incubated overnight at 4 °C. The plate was then washed twice with RIA buffer (7.5 pH) and incubated overnight at 4 °C with 200 μL of the anti-body serum diluted at 1:20,000 for CORT and 1:800 for DHEA-S. The cross-reactivities of the anti-CORT antibody with other steroids were as follows: cortisol, 100%; cortisone, 4.3%; corticosterone, 2.8%; 11-deoxycorticosterone, 0.7%; 17-hydroxyprogesterone, 0.6%; dexamethasone, 0.1%; progesterone, 17-hydroxypregnenolone, DHEA-S, androsterone sulphate, pregnenolone, <0.01%. The cross-reactivities of the anti-DHEA-S antibody with other steroids were as follows: DHEA-S, 100%; androstenedione, 0.2%; DHEA, <0.01%; androsterone, <0.01%; testosterone, <0.01%. After washing the plate with RIA buffer, the standards (5–200 pg/well), the quality control extract, the test extracts and the tracer (hydrocortisone {cortisol [1,2,6,7-3H (N)]-}, DHEA-S [1,2,6,7-3H (N)] were added, and the plate was incubated overnight at 4 °C. The bound hormone was separated from the free hormone by decanting and washing the wells in the RIA buffer. After the addition of 200 μL of scintillation cocktail, the plate was counted on a β-counter (Top-Count, Perkin-Elmer Life Science, Boston, MA, USA).

The intra- and inter-assay coefficients of variation were 3.6% and 9.8% for CORT and 3.6% and 12.7% for DHEA-S. The sensitivities of the assays were 24.6 pg/mL for CORT and 15.8 pg/mL for DHEA-S.

### 2.4. Statistical Analysis

Pearson test was used to correlate foal’s and mare’s hair hormone concentrations within the same group and with mare’s age, parity, gestation length and foal’s weight.

In order to obtain a correct approach to study the relative abundance of the two hormones and determine which is preferentially produced, the following ratios were calculated: the ratio between CORT and DHEA-S concentrations in foals, ratio between CORT and DHEA-S concentrations in mares, ratio between the foal and mare CORT concentrations, ratio between foal and mare DHEA-S concentrations and ratio between foal and mare CORT/DHEA-S ratios.

Independent samples *t*-test was used to evaluate: differences in foals’ hormone concentrations in relation to sex and outcome (survivors vs. non-survivors) and those of the mare in relation to fetal sex; differences in hormone concentrations and their ratio between H and S groups (healthy vs. sick; study 1); differences in hair hormone concentrations between HH and HB groups (hospital vs. breeding farm; study 2).

Data were expressed as mean ± SD, min-max. A *p* < 0.05 was considered statistically significant. All statistical analyses were carried out using SPSS software (Statistic version 25 IBM, SPSS Inc., Chicago, IL, USA).

## 3. Results

### 3.1. Clinical Data

In study 1, the 107 animal pairs were divided into two groups: the one with healthy foals (group H; *n* = 56 pairs) and the one with sick foals (group S; *n* = 51 pairs). The clinical data collected from mares and foals of the two groups are shown in Table 1.

In group S, 38/51 foals (74.5%) were born from apparently normal pregnancy and 13/51 foals (25.5%) were born from high-risk pregnancy. Foals’ diagnoses included Neonatal Syndrome (15/51; 29.4%), Hypoxic-Ischemic Encephalopathy (14/51; 27.5%), prematurity (5/51; 9.8%), dysmaturity (5/51; 9.8%), stillbirth (2/51; 3.9%) and flexural deformity (10/51; 19.6%). In mares with high-risk pregnancy, the condition was associated with placentitis (7/13; 53.9%), placental edema (2/13; 15.4%), twin pregnancy (1/13; 7.7%), colic syndrome (2/13; 15.4%) and piroplasmosis (1/13; 7.7%). Ten/13 mares (76.9%) received treatment for the underlying condition.

In study 2, out of the 56 animal pairs of group H, 30 mares delivered at the hospital (group HH) and 26 at the breeding farm (group HB).

### 3.2. CORT and DHEA-S Concentrations

No differences in the concentration of hormones extracted from the trichological matrix were found in the entire population between colts and fillies, between mares with male and female fetuses, and between surviving and non-surviving foals. Considering all animals, no correlations were demonstrated between the foal and mare CORT and DHEA-S concentrations and mare’s age and parity, gestation length, and foal’s weight.

In study 1, CORT and DHEA-S concentrations in foals and mares and their resulting ratios of H and S groups are shown in Table 2. The frequency distribution graphs of the foal CORT/DHEA-S ratio, mare CORT/DHEA-S ratio and foal CORT/DHEA-S—mare CORT/DHEA-S ratio are shown in Figure 1, Figure 2 and Figure 3, respectively. In group H, weak positive correlations were found between foal CORT and DHEA-S concentrations (*p* = 0.018; r = 0.282), mare CORT and DHEA-S concentrations (*p* = 0.006; r = 0.361) and CORT concentrations in foals and mares (*p* = 0.019; r = 0.312). In group S, a strong positive correlation was found between foal CORT and DHEA-S concentrations (*p* < 0.001; r = 0.44), whereas weak positive correlations were found between mare CORT and DHEA-S concentrations (*p* = 0.027; r = 0.271) and CORT concentrations in foals and mares (*p* = 0.006; r = 0.349). In foals, DHEA-S concentrations were higher in group S than in group H (*p* = 0.033), whereas CORT/DHEA-S ratio was lower in group S than in group H (*p* < 0.001).

In study 2, CORT and DHEA-S concentrations in foals and mares and their resulting ratios of HH and HB groups are shown in Table 3; no significant differences were found.

## 4. Discussion

In this study, the content of CORT and DHEA-S in the trichological matrix has been evaluated for the first time in mares and their foals affected by the neonatal disease. The dosage of CORT from foal hair has already been performed in previous studies, but only in healthy subjects born from normal pregnancy [11,12]. In contrast, the dosage of DHEA/DHEA-S has been performed in a few recent studies but only in adult horses [16,17,31].

Our first hypothesis that glucocorticoid concentrations detected individually in foal’s hair are influenced by the endocrine setting of the respective mare remains under investigation. In the present study, a correlation between foal and mare hair CORT concentrations exists in healthy and sick foals, which may imply that prepartum CORT deposition in fetal hair occurs depending on changes in maternal plasma CORT concentration. In humans, pregnancy-specific stress in the third trimester was positively related to neonatal hair CORT [32]. No relationship was found in a study that compared hair CORT concentrations in mothers and premature infants immediately after birth [33]. Regarding the equine species, foals born from mares with systemic illness or placental dysfunction show evidence of increased adrenocortical function [34], involving an increase in CORT concentration. Initially, some authors have observed a synchronization between fetal and maternal plasma ACTH [35,36,37]; furthermore, sustained increases in maternal plasma CORT following repeated treatment with exogenous ACTH induce precocious fetal maturation and then cause premature delivery of viable foals [38]. Theoretically, there is little transplacental transfer of maternal CORT to the fetus throughout gestation because the placenta possesses 11β-hydroxysteroid dehydrogenase (11β-HSD), which converts CORT into inactive cortisone, protecting the fetus from fluctuations in maternal CORT concentrations [35,39]. It has not yet been established if the unidirectional type II form of 11β-HSD predominates in the equine placenta, as it does in other species [40]. A more recent study, based on the partial passage of 14C-labeled CORT infused into the fetus and diffused into the maternal circulation during the last weeks of gestation, has supported a possible placental transfer of endogenous CORT from the fetus to the mare [41], which may increase with changes in placental structure and steroid metabolism in the prepartum period. Regarding changes in maternal salivary and plasma CORT concentrations before and after parturition, the authors hypothesized that the induction of parturition by the fetus might be comparable to a chronic stress response that triggers prepartum increases in CORT [41]. The increase in CORT concentrations in the antepartum mare suggests that CORT in the maternal circulation could in part be of fetal origin. Regardless, further studies are needed to identify the precise mechanism of CORT transfer from maternal to fetal circulation or vice versa and thus explain the correlation in the hair CORT deposition of the mare-foal pairs.

A correlation was also found between CORT and DHEA-S concentrations within the same subject, in both foals and mares of each group. These findings demonstrate the close relationship between the two hormones in the regulation of the HPA axis. Stimulation of the HPA axis and increased CORT concentrations in fetal plasma [7] and hair [11] have been associated with neonatal maturity, viability, and adaptation to extra-uterine life in the equine species. Little is known about plasma DHEA concentrations in foals [9], and to the authors’ knowledge, this is the first study evaluating DHEA-S concentrations in the hair of newborn foals. The CORT/DHEA-S ratio combines the individual concentrations of the two glucocorticoids into a single piece of information about HPA axis activity. Thus, by comparing the frequency distribution graph of the CORT/DHEA-S ratio in healthy and sick foals (Figure 1), it was possible to observe that higher values of the ratio in healthy foals were more frequent than in sick ones. The frequency distribution graph of the CORT/DHEA-S ratio in mares (Figure 2) denotes the opposite: mares that foaled sick foals showed higher ratios than those that foaled healthy ones. From this scenario, it cannot be excluded that the mare may influence the reprogramming of fetal HPA axis activity when subjected to chronic pathological conditions. Increased CORT concentrations and CORT/DHEA-S ratio in pregnant mares could imply an impaired fetal development and thus the birth of foals with a reduced CORT/DHEA-S ratio. In addition, a lower ratio between foal and mare CORT/DHEA-S ratios (Figure 3) was observed in sick foals; the “double ratio” could also be informative of HPA axis activity.

Our second hypothesis that foals classified as sick at birth have different hormone concentrations than healthy ones was confirmed. In sick foals, DHEA-S concentrations are higher than in healthy ones, and therefore, a decrease in the ratio of CORT to DHEA-S takes place. DHEA is synthesized in steroidogenic tissues, such as adrenals, gonads and placenta, and in the nervous system from pregnenolone [42]. The equine fetus produces a considerable amount of DHEA that is probably used by the allantochorion to synthesize estrogens [43]. However, a considerable amount of DHEA, and its sulfated metabolite DHEA-S, escapes the placenta, as their concentrations in the maternal circulation are elevated until mid-pregnancy [44,45]. High DHEA-S concentration in sick foals could be a positive finding, as this steroid seems to contribute to animal health conditions and wellbeing. Indeed, DHEA and DHEA-S would be important as neuroactive steroids in the regulation of neural function, such as neuroprotection, neurogenesis, neuronal growth, and differentiation. Moreover, these hormones would influence catecholamine synthesis and secretion, have antioxidant and anti-inflammatory action and antagonize glucocorticoid effects [46]. More than half of the sick foals were affected by Hypoxic-Ischemic Encephalopathy/Neonatal Syndrome. Thus the accumulation of DHEA-S in the trichological matrix may be related to its increased production to comply with its neuroprotective role. The CORT/DHEA ratio may serve as a diagnostic or prognostic tool for some species in terms of physical health and a potential biomarker of resilience and allostatic load [47]. In the bovine species, there is evidence that, compared to healthy cows, lame cows had impaired CORT/DHEA ratios, exhibited less eating and ruminating, and performed more self-grooming, suggesting that the CORT/DHEA ratio may serve as a biomarker of inflammatory foot lesions [48]. Moreover, circulating DHEA was higher also in cows with metritis, associated with a lower CORT/DHEA ratio [49]. The authors concluded that DHEA and CORT/DHEA ratio could represent an anti-inflammatory signal during prolonged inflammation and a putative prognostic biomarker for evaluating disease severity. Increased hair DHEA-S concentrations and decreased CORT/DHEA-S ratio appear to be potential signs of resilience even in sick foals and deserve further attention.

Some studies have focused on hair CORT concentrations in relation to temporary relocation [50], management regimes [51,52] and welfare [31]. Of interest, relocated horses exhibited elevated hair CORT concentrations compared with control horses after the relocation period, suggesting a change in their welfare status, probably related to the sudden change in their surrounding conditions [50]. Our third hypothesis that hormone concentrations detected in the hair of mares hospitalized for attended parturition reflect different HPA axis activity from those delivered at the breeding farm was not confirmed. In companion and farm animals, it has been shown that housing conditions affect HPA axis activity and thus hair CORT concentrations. Solitary housing of dogs decreased hair CORT concentration compared with dogs in multi-dog households [53] but increased hair CORT concentration compared with paired housing [54]. In pigs, housing in barren conditions caused significantly higher hair CORT concentrations compared with housing in enriched pens [55]. In beef cattle, a substantial reduction in stocking density resulted in significantly increased CORT concentrations in tail switch hair [56]. In contrast, minor differences in stocking density of cattle did not affect hair CORT concentration [57]. The relocation of rabbits [58] and cows [59] from their habitual environment are also stressors that can cause an increase in hair CORT concentration. Interestingly, in the study of Peric et al. [58], relocation induced an increase in hair CORT concentration and the change of employees in the facility. In the only study conducted on adult horses undergoing temporary relocation, stallions exhibited elevated hair CORT concentrations compared with control horses, despite the small number of samples [50]. In the present study, mares were transferred from the original breeding farm to the hospital at about 310 days of pregnancy. They were housed in separate boxes during the hospitalization period and could graze during the day in individual paddocks 20 × 20 m wide. Moreover, they were clinically evaluated twice a day and by ultrasonography every 10 days until parturition. Despite the relocation and the stressful procedures, the hospitalization in our facility was not accompanied by fluctuations in hair CORT concentrations, in contrast to the previous study by Gardela et al. [50]. On the one hand, these results could be related to a too-brief period of hospitalization to cause significant changes in steroid deposition in the mare’s hair. On the other hand, the wide range of different stressors experienced by the mares was probably well balanced by the high level of care in terms of feeding and nutritional status, bedding and rest quality, socialization, environmental enrichment, the timing of procedures, and access to outdoor provided to the animals at the hospital.

From a clinical perspective, the main limitation of the study design should be noted. The population of sick foals was not perfectly homogeneous. In all, the diagnoses were related to intrauterine life span dysfunction, but a diagnosis of placental dysfunction or systemic illness was not reached in most of their mares. In contrast, the few mares with a diagnosis of high-risk pregnancies received treatment. Nevertheless, the population offered a pure and spontaneous model of equine neonatal disease. These results provide a starting point to better understand the role of steroids in the trichological matrix in the equine perinatal period. However, they need to be confirmed by further studies in a larger and more homogeneous population of sick foals.

## 5. Conclusions

In conclusion, this study’s findings provide evidence that the HPA axis in sick foals can be assessed through hair analysis and how fine connections exist between foal, mare, and their clinical conditions. Since the prenatal HPA axis activity is pivotal to the final fetal maturation and to its adaptation to extra-uterine life, hair glucocorticoid concentrations deserve to be further investigated as potential biomarkers of prenatal diseases, as well as to evaluate resilience and allostatic load in sick foals.

Although the specific role of CORT and DHEA-S in mares and foals under pathological conditions remains under investigation, this study opens up the use of neonatal hair segments as an attractive non-invasive retrospective calendar for the assessment of prenatal HPA axis activity in the equine species.

## Figures and Tables

**Figure 1 animals-12-01266-f001:**
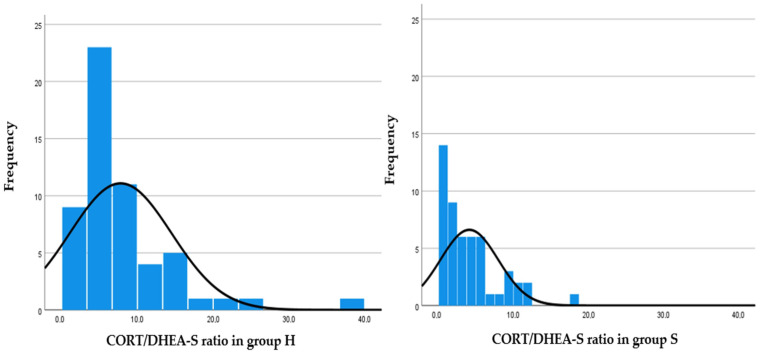
Frequency distribution graphs representing CORT/DHEA-S ratio in healthy (group H; left) and sick (group S; right) foals.

**Figure 2 animals-12-01266-f002:**
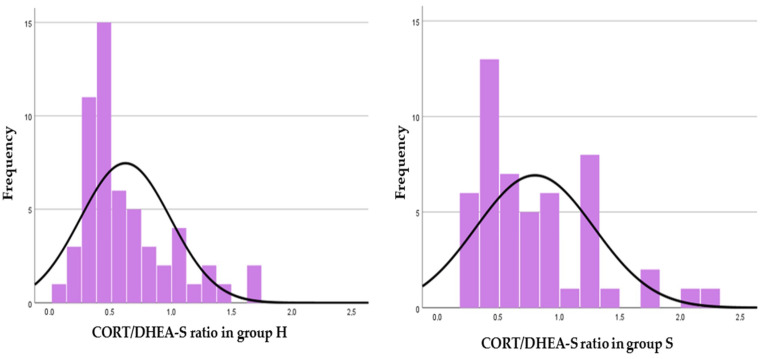
Frequency distribution graphs representing CORT/DHEA-S ratio in mares of the two groups (healthy foals—H, left; sick foals—S, right).

**Figure 3 animals-12-01266-f003:**
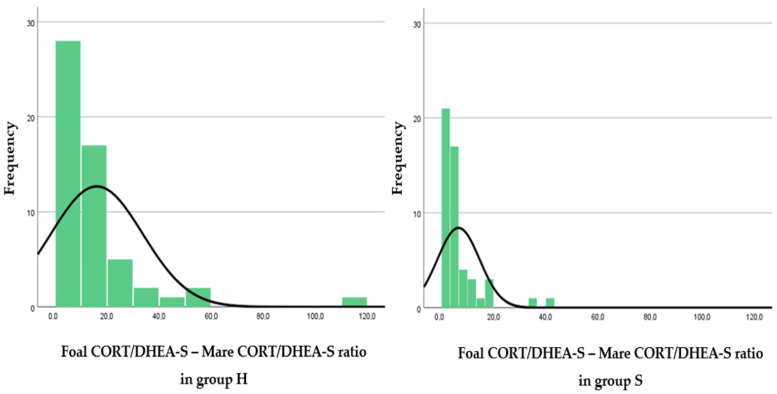
Frequency distribution graphs representing the ratio between foal CORT/DHEA-S ratio and mare CORT/DHEA-S ratio of the two groups (healthy foals—H, left; sick foals—S, right).

**Table 1 animals-12-01266-t001:** Study 1, clinical data collected for mares and foals of the two groups (Heathy—H and Sick—S). Data are expressed as mean ± standard deviation (min-max). Stb = Standardbred; Other = other breeds; N = normal pregnancy; HR = high-risk pregnancy; AN = apparently normal pregnancy; Sv = survived to hospital discharge; NSv = not survived.

	Breed	Age (Years)	Parity (n)	Type of Pregnancy (N/HR/AN)	Gestation Length (Days)
Group H (*n* = 56)	Stb *n* = 34 Other *n* = 22	11 ± 5 (4–24)	4 ± 3 (1–14)	N	340 ± 11 (321–371)
Group S (*n* = 51)	Stb *n* = 31 Other *n* = 20	11 ± 4 (4–21)	3 ± 3 (1–10)	HR *n* = 13 AN *n* = 38	332 ± 16 (281–360)
	Sex	Weight (kg)	Apgar score	Age at admission (days)	Outcome
Group H (*n* = 56)	Male *n* = 23 Female *n* = 33	50 ± 8 (38–73)	10 ± 1 (8–10)	3 ± 4 (0–13)	Sv
Group S (*n* = 51)	Male *n* = 27 Female *n* = 24	45 ± 8 (25–62)	8 ± 2 (0–10)	2 ± 3 (0–13)	Sv *n* = 44 NSv *n* = 7

**Table 2 animals-12-01266-t002:** Results of study 1 in the two groups (Healthy—H and Sick—S). Data are expressed as mean ± standard deviation (min-max). * Superscript asterisk in a row indicates a significant difference (*p* < 0.05) among groups with independent samples *t*-test.

	Group H (*n* = 56 Pairs; 112 Animals)	Group S (*n* = 51 Pairs; 102 Animals)
Foal CORT (pg/mg)	71.5 ± 38.2 (27.7–258.9)	62.2 ± 36.1 (6.3–200.7)
Foal DHEA-S (pg/mg)	19.2 ± 44.0 (2.2–333.9)	43.1 ± 69.0 (4.5–285.3) *
Foal CORT/DHEA-S ratio	7.8 ± 6.7 (0.3–38.9)	4.1 ± 3.8 (0.2–17.7) *
Mare CORT (pg/mg)	5.4 ± 2.4 (2.6–18.5)	6.6 ± 4.9 (2.1–28.2)
Mare DHEA-S (pg/mg)	12.1 ± 14.1 (2.6–105.7)	9.0 ± 3.9 (3.1–20.9)
Mare CORT/DHEA-S ratio	0.6 ± 0.4 (0.1–1.7)	0.8 ± 0.5 (0.2–2.3) *
Foal/Mare CORT ratio	14.9 ± 10.4 (3.2–76.2)	11.6 ± 7.3 (2.0–28.2)
Foal/Mare DHEA-S ratio	1.8 ± 2.9 (0.1–22.3)	5.1 ± 7.7 (0.4–32.7) *
Foal CORT/DHEA-S - Mare CORT/DHEA-S ratio	16.0 ± 17.6 (1.0–113.1)	6.5 ± 8.1 (0.4–42.0)

**Table 3 animals-12-01266-t003:** Results of study 2 in group Healthy (H) based on housing location at foaling (hospital—group, HH vs. breeding farm—group, HB). Data are expressed as mean ± standard deviation (min-max).

	Group HH: Hospital (*n* = 30 Pairs; 60 Animals)	Group HB: Breeding Farm (*n* = 26 Pairs; 52 Animals)
Foal CORT (pg/mg)	71.9 ± 39.4 (43.1–258.9)	71.0 ± 37.5 (27.7–201.2)
Foal DHEA-S (pg/mg)	12.0 ± 7.4 (2.2–33.3)	27.5 ± 63.7 (2.7–333.9)
Foal CORT/DHEA-S ratio	9.0 ± 8.3 (2.0–39.5)	6.3 ± 4.2 (0.3–15.5)
Mare CORT (pg/mg)	5.4 ± 2.8 (2.8–18.5)	5.4 ± 2.0 (2.6–11.3)
Mare DHEA-S (pg/mg)	9.7 ± 3.7 (2.6–17.5)	15.0 ± 20.1 (2.8–105.7)
Mare CORT/DHEA-S ratio	0.7 ± 0.4 (0.2–1.7)	0.6 ± 0.3 (0.1–1.5)
Foal/Mare CORT ratio	15.4 ± 12.3 (3.2–76.2)	14.2 ± 7.9 (5.1–42.8)
Foal/Mare DHEA-S ratio	1.3 ± 0.8 (0.3–4.1)	2.3 ± 4.2 (0.1–22.3)

## Data Availability

Not applicable.

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
