# Peer review of "Hair Cortisol and DHEA-S in Foals and Mares as a Retrospective Picture of Feto-Maternal Relationship under Physiological and Pathological Conditions"

_animals, 2022, doi:10.3390/ani12101266_

Round 1

Reviewer 1 Report

The present study would have as general objective the analysis of cortisol/DHEA concentrations in the hair of newborn foals in order to understand, retrospectively (last third of pregnancy 270 days), fetal maternal relationship under physiological and pathological conditions), being useful in the future as a biomarker of chronic stress in horses, in the final phase of pregnancy.

Abstract: I believe that some speculative aspects described in the discussion (387-392) were proposed as a cause-effect in the abstract of the article. I'd take it off. In order to make the reading more attractive, I would highlight in the abstract the possibility of being a biomarker of chronic stress in the final third of pregnancy and may be in the future as the increase in concentrations of DHEA-S (422) and the decrease in the CORT/DHEA-S ratio seem to be potential signs of resilience even in sick foals, and deserve more attention. Allostasis, would further explore the biomarker aspect  in the evaluation of prenatal HPA axis activity in the neonatology of equine species.

I would also discuss a little more in relation to other forms of chronic stress detection as proposed by Nage's work (Nagel, C.; Erber, R.; Bergmaier, C.; Wulf, M.; Aurich, J.; Möstl, E.; Aurich, C. Cortisol and progestin release, heart rate and 586 heart rate variability in the pregnant and postpartum mare, fetus and newborn foal. Theriogenology 2012, 78, 759-767 ).

The cortisol/DHEA ratio has been considered more informative of the relative action of the two steroids in the brain than their isolated values) lower ratio due to the increase in DHEA (PROTECTIVE ACTION, COUNTERREGULATORY OF CORTISOL).  In human studies,  suggest that DHEA reduces glucocorticoide receptor levels, inhibits excitatory neurotransmitters and promotes increased neurotrophins. 

In keywords:  Would remove the words cortisol and dehydroepiandrosterone are already in the title of the article I suggest include allostasis and biomarks. 

Results:

In table 2 of results, I would suggest placing foal/maternal close. Cortisol foal/mother; Foal/mother DHEAs in all parameters analysed in healthy/ill patients. So that it would be more easily visualized.

Graphics of similar sizes would be better.

Correct Group of the ratio cortisol/DHEA mothers sick group because it is gropu FIGURE

I believe that in relation to the three hypotheses of the study  the authors could make clearer the answers to the hypotheses formulated in the study

1)The hormonal concentrations detected individually in the foal are influenced by the endocrine adjustment of the respective mare. Proposed in the discussion: Theoretically, there is little transplacental transfer of maternal CORT to the fetus throughout gestation, because the placenta possesses 11β-hydroxysteroid dehydrogenase (11β-HSD) 364 which converts CORT in inactive cortisone, protecting the fetus from fluctuations in maternal CORT concentrations . Regardless, further studies are needed to identify the precise mechanism of CORT transfer from maternal to fetal circulation or vice versa, and thus explain the correlation in the hair CORT deposition of the mare-foal pairs. 

2)Foals classified as sick at birth have different hormone concentrations than healthy foals: The cortisol/DHEA ratio has been considered more informative of the relative action of the two steroids in the brain than their isolated values) lower ratio due to the increase in DHEA (PROTECTIVE ACTION, COUNTERREGULATORY OF CORTISOL), this is important discussion. Yes

3)Hormonal concentrations detected in mares hospitalized for assisted parturition reflect different HPA axis activities of those in breeding farms. No

Author Response

Reviewer #1

The present study would have as general objective the analysis of cortisol/DHEA concentrations in the hair of newborn foals in order to understand, retrospectively (last third of pregnancy 270 days), fetal maternal relationship under physiological and pathological conditions), being useful in the future as a biomarker of chronic stress in horses, in the final phase of pregnancy.

Abstract: I believe that some speculative aspects described in the discussion (387-392) were proposed as a cause-effect in the abstract of the article. I'd take it off. In order to make the reading more attractive, I would highlight in the abstract the possibility of being a biomarker of chronic stress in the final third of pregnancy and may be in the future as the increase in concentrations of DHEA-S (422) and the decrease in the CORT/DHEA-S ratio seem to be potential signs of resilience even in sick foals, and deserve more attention. Allostasis, would further explore the biomarker aspect in the evaluation of prenatal HPA axis activity in the neonatology of equine species.

Thanks for the suggestion. The abstract has been partially modified in light of your suggestions to get a more attractive and clear one. Please see lines #35-51.

I would also discuss a little more in relation to other forms of chronic stress detection as proposed by Nagel's work (Nagel, C.; Erber, R.; Bergmaier, C.; Wulf, M.; Aurich, J.; Möstl, E.; Aurich, C. Cortisol and progestin release, heart rate and heart rate variability in the pregnant and postpartum mare, fetus and newborn foal. Theriogenology 2012, 78, 759-767).

The authors consider Nagel's work to be of a high standard. We think it may be important to give more space to its important findings in the discussion section, although salivary and plasma matrixes are very different from the trichological ones. Please see lines #394-405.

The cortisol/DHEA ratio has been considered more informative of the relative action of the two steroids in the brain than their isolated values) lower ratio due to the increase in DHEA (PROTECTIVE ACTION, COUNTERREGULATORY OF CORTISOL).  In human studies, suggest that DHEA reduces glucocorticoide receptor levels, inhibits excitatory neurotransmitters and promotes increased neurotrophins.

We completely agree with the concept of CORT/DHEA-S ratio. DHEA and DHEA-S would be important as neuroactive steroids in the regulation of neural function, such as neuroprotection, neurogenesis, neuronal growth, and differentiation. To note, more than half of the sick foals were affected by Hypoxic-Ischemic Encephalopathy/Neonatal Syndrome, characterized by neuronal failure resulting from hypoxia and ischemia of the brain, and thus the accumulation of DHEA-S in the trichological matrix may be related to its increased production to comply with its neuroprotective role. Please see lines #430-456.

In keywords:  Would remove the words cortisol and dehydroepiandrosterone are already in the title of the article I suggest include allostasis and biomarks.

Thanks for the suggestion, we will definitely apply it. Please see lines #55-56.

Results:

In table 2 of results, I would suggest placing foal/maternal close. Cortisol foal/mother; Foal/mother DHEAs in all parameters analysed in healthy/ill patients. So that it would be more easily visualized.

Thanks for the suggestion. If possible, the authors would prefer to maintain the following logical order: foal (CORT, DHEA-S and ratio), mare (CORT, DHEA-S and ratio) and then ratios between foal and mare hair hormone concentrations. We hope the reviewer will accept this form. Thanks.

Graphics of similar sizes would be better.

Absolutely yes, we will edit them, thanks.

Correct Group of the ratio cortisol/DHEA mothers sick group because it is gropu FIGURE

Apologize for the error, thanks. Please see Figure 1 and 2.

I believe that in relation to the three hypotheses of the study the authors could make clearer the answers to the hypotheses formulated in the study

1) The hormonal concentrations detected individually in the foal are influenced by the endocrine adjustment of the respective mare. Proposed in the discussion: Theoretically, there is little transplacental transfer of maternal CORT to the fetus throughout gestation, because the placenta possesses 11β-hydroxysteroid dehydrogenase (11β-HSD) 364 which converts CORT in inactive cortisone, protecting the fetus from fluctuations in maternal CORT concentrations . Regardless, further studies are needed to identify the precise mechanism of CORT transfer from maternal to fetal circulation or vice versa, and thus explain the correlation in the hair CORT deposition of the mare-foal pairs. 

2) Foals classified as sick at birth have different hormone concentrations than healthy foals: The cortisol/DHEA ratio has been considered more informative of the relative action of the two steroids in the brain than their isolated values) lower ratio due to the increase in DHEA (PROTECTIVE ACTION, COUNTERREGULATORY OF CORTISOL), this is important discussion. Yes

3) Hormonal concentrations detected in mares hospitalized for assisted parturition reflect different HPA axis activities of those in breeding farms. No

Thanks for the suggestion. We have made the answers to the three hypotheses clearer.

1) Please see lines #375-377, and argumentation #377-405.

2) Please see lines #428-430, and argumentation #430-456.

3) Please see lines #461-464, and argumentation #464-489.

Reviewer 2 Report

The authors report on hair values of CORT and DHEA-S in foals and mares. The manuscript offers a valuable data to the veterinary literature, is well organized and provides conclusions that are supported by the data provided. The research and data obtained are interesting, primarily due to the small number of reports regarding hormones analysis in horses hair.

I have only few comments on the article:

Line 65-68: very long sentence, I would recommend splitting it into two

Line 79: “hair hormones” - sounds quite unfortunate

Line 99-104: again a very long sentence. The ":" and the connection between the part before and after it are not fully understood. Please rephrase.

Line 104-107: please clarify. It is an alternative, but only after a certain period of pregnancy, as the author mentioned earlier in the introduction. We will not replace the classic examination if the fetus still does not have hair.

Line 111-112: “some” ? – “selected” would be better word.

Line 185: “exitus”?

Line 187: please explain why this area was selected. Earlier, the article did not specify which hair was tested - that's why not a mane?

Line 260-263: the authors chose multiple relations for evaluation, however early did not explain why, as well as it does not appear in the methods. I would advise you to present these relations differently in the text, because in the continuous text it is very difficult to catch what was compared to what or what was subtracted from what (and why?).

Line 344-350: in this part of the discussion, I was unable to tell why this result was obtained.

Table 2 and 3: Results of the study…? Rewrite the titles.

Figure 2: “gropu” should be “group”

Author Response

Reviewer #2

The authors report on hair values of CORT and DHEA-S in foals and mares. The manuscript offers a valuable data to the veterinary literature, is well organized and provides conclusions that are supported by the data provided. The research and data obtained are interesting, primarily due to the small number of reports regarding hormones analysis in horses hair.

 I have only few comments on the article:

Line 65-68: very long sentence, I would recommend splitting it into two

Done. Thanks for the suggestion. Please see lines #73-76.

Line 79: “hair hormones” - sounds quite unfortunate

We are sorry if it sounds bad but “hair hormones” is the term used in all publications related to hormone levels measured in the trichological matrix. We hope the reviewer will accept this form. Thanks.

Line 99-104: again a very long sentence. The ":" and the connection between the part before and after it are not fully understood. Please rephrase.

Done. Thanks for the suggestion. Please see lines #107-113.

Line 104-107: please clarify. It is an alternative, but only after a certain period of pregnancy, as the author mentioned earlier in the introduction. We will not replace the classic examination if the fetus still does not have hair.

Thanks for the suggestion. It is important to be precise, which is why we have replaced “prenatal period” with the more accurate term “final third of pregnancy”. Please see line #114.

Line 111-112: “some”? – “selected” would be better word.

We agree, thank you for the suggestion. “Selected” seems more appropriate than “some”. See lines #120-121.

Line 185: “exitus”?

“Exitus” is the term used in our clinical practice to express the patient's outcome relative to survival. However, we have changed it to the more commonly used term “outcome” throughout the manuscript (in the present study survived or not to hospital discharge as specified in the text). Thanks for the suggestion. Please see line#122 line#194 line#240 #Table 1.

Line 187: please explain why this area was selected. Earlier, the article did not specify which hair was tested - that's why not a mane?

In previous studies, conducted by other authors, the most disparate body regions were sampled: neck, withers, tail and mane. The authors of the present study standardized the sternal region in the foal and the dorsal cervical region in the mare. Firstly, in agreement with the owners: the shaving of the cervical area under the mane and the sternal region is not visible and foals can be sold at auction without visible shaving marks in the coat. Secondly, the sternum in foals is a region less susceptible to weather exposure, fecal contamination, and grooming, and has optimal hair growth rates and skin blood flow.

Line 260-263: the authors chose multiple relations for evaluation, however early did not explain why, as well as it does not appear in the methods. I would advise you to present these relations differently in the text, because in the continuous text it is very difficult to catch what was compared to what or what was subtracted from what (and why?).

Thanks for the suggestion. We have added a brief paragraph in the M&M section that explains the calculation of ratios. Please see lines #233-238.

In addition, the text was made easier to read, when ratios were presented. Please see lines #275-278 and lines #351-353.

Line 344-350: in this part of the discussion, I was unable to tell why this result was obtained.

We agree with the reviewer: this part of the discussion is out of context and does not bring new elements to the discussion. We have decided to remove it. We hope the reviewer will accept the decision. Thanks. Please see lines #368-374.

Table 2 and 3: Results of the study…? Rewrite the titles.

Thanks for the suggestion. The titles of Table 2 and 3 have been reworded and now appear easier to read. Please see lines #288-294 for Table 1 and lines #355-360 for Table 3.

Figure 2: “gropu” should be “group”

Apologize for the error, thanks. Please see Figure 2 and 3.

Reviewer 3 Report

In my opinion the manuscript can be accepted in the present form.

Author Response

Reviewer #3
In my opinion the manuscript can be accepted in the present form.
The authors wish to thank the reviewer for approval.